# The Mechanism of Mineral Dissolution on the Development of Red-Bed Landslides in the Wudongde Reservoir Region

Chenglin Ye [1], Jingmin Liu [2], Yan Shi [3], Siyuan Zhao [1], Hua Li [1] and Jianhui Deng [1,*]

[1] College of Water Resource, Hydropower Sichuan University, Chengdu 610065, China; yechenglin@stu.scu.edu.cn (C.Y.); zhaosiyuan@scu.edu.cn (S.Z.); huali@scu.edu.cn (H.L.)
[2] PowerChina Huadong Engineering Corporation Limited, Hangzhou 311122, China; liu_jm6@hdec.com
[3] Three Gorges Geotechnical Engineering Co., Ltd., Wuhan 430014, China; 15271901004@163.com
[*] Correspondence: jhdeng@scu.edu.cn; Tel.: +86-13-980-959-251

**Abstract:** The Wudongde reservoir region exhibits a notable prevalence of landslides within the red-bed reservoir stratum. The red bed is a clastic sedimentary rock layer dominated by red continental deposits. It is mainly composed of sandstone, mudstone, and siltstone. The lithology is diverse and uneven. In this study, we delve into the impact of mineral dissolution on the development of red-bed landslides in the reservoir area by utilizing the Xiaochatou landslide as a representative case study. Considering the inherent susceptibility of red-bed formations to erosion, collapse, and softening when exposed to water, an investigation was conducted to examine the consequences of mineral dissolution on landslides occurring in these areas. We conducted a mineral analysis and an identification of rock samples from the Xiaochatou landslide site, revealing alternating layers of sandstone and mudstone. Sandstone and conglomerate specimens were immersed in deionized water, and advanced techniques such as scanning electron microscopy (SEM), ion chromatography (IC), and inductively coupled plasma (ICP) analysis were used to examine the effects of water immersion. We also employed the hydrogeochemical simulation software PHREEQC to understand the dissolution mechanism of gypsum during soaking. Our findings reveal that sandstone and conglomerates harbor a notable quantity of gypsum, which readily dissolves in water. Prolonged immersion leads to erosion cavities within the sandstone, thereby augmenting its permeability. The concentration of $SO_4^{2-}$ ions in the soaking solution emerges as the highest, followed by $Ca^{2+}$ and $Na^+$. The notable significance is the dissolution of gypsum, whose intricate mechanism is contingent upon diverse environmental conditions. Variations in ion concentration profoundly influence the saturation index (SI) value, with the pH value playing a crucial role in shifting the reaction equilibrium. Regarding the deformation mode of the landslide, it manifests as a combination of sliding compression and tension cracking. The fracture surface of the landslide assumes a step-like configuration. As the deformation progresses, the mudstone layer takes control over the sliding process, causing the sandstone to develop internal narrow-top and wide-bottom cracks, which propagate upward until the stability of the slope rock mass is compromised, resulting in its rupture. In this manuscript, we delve into the dissolution traits of red-bed soft rock in the Wudongde reservoir area, using a landslide case as a reference. We simulate this rock's dissolution under environmental water influences, examining its interaction with diverse water types through rigorous experiments and simulations. This study's importance lies in its potential to shed light on the crucial engineering characteristics of red-bed soft rock.

**Keywords:** red-bed landslide; mineral composition; microstructure; hydrochemistry simulation; mechanism of water–rock reaction; scanning electron microscopy (SEM); siliciclastic rocks and evaporites

## 1. Introduction

The red beds in China can be categorized into several types, including the southwest red beds, dominated by the Sichuan–Yunnan red beds; northwest red beds, dominated

by the Gansu red beds; central and southeast red beds; and other red beds, dominated by the Tibetan red beds [1–3]. The red layer exhibits a degree of concealment, with its distribution dictated by the fractured rock structure of the fault zone. This rock mass demonstrates a high hydrophilicity, leading to potential engineering geological issues such as seepage, softening, and disintegration upon water exposure. Typically, red-bed landslides are triggered by rainfall or reservoir water level fluctuations [4].

Under the influence of groundwater seepage, weak interlayers are water-softened, evolving into slip zone soil. The presence of this unique water–rock interaction is intimately tied to the rock mass type and structure, as well as the content of hydrophilic minerals, such as clay minerals.

The red bed in the Jinsha River, an essential part of the Sichuan–Yunnan red bed strata, has garnered the attention of researchers due to the construction of cascade hydropower stations in the river's lower reaches. Due to its unique engineering properties, the red-bed area often experiences geological disasters like landslides. The Wudongde reservoir area, particularly, is characterized by a high incidence of red-bed landslides and is associated with a large-scale landslide river-blocking event in history. Therefore, it is crucial to study these landslides [5,6].

The Sichuan Basin and the Three Gorges Reservoir area in China are characterized by the prevalence of red-bed strata, commonly called "landslide-prone strata" [7]. Red-bed strata are terrestrial sedimentary formations characterized by alternating layers of red mudstone and sandstone [8]. These landslides are commonly initiated by precipitation and occur along the planes of stratification. A notable instance is the torrential rainfall experienced on July 9 and 10, 1989, surpassing 200 mm across the majority of the Sichuan Basin. This event instigated over 10,000 landslides, predominantly comprising bedding plane landslides [9]. Under the influence of groundwater infiltration, the presence of weak interlayers undergoes a process of softening upon encountering water, subsequently developing into slip zones [10]. The underlying reasons for this phenomenon, which involves unique hydro-mechanical interactions, are closely associated with the type of rock mass, its structural characteristics, and the content of hydrophilic minerals such as clay minerals [11]. The dissolution characteristics of gypsum-bearing red layers have received significant attention. When these red layers come into contact with water, gypsum undergoes extensive dissolution, resulting in mineral loss within the rock mass. This process leads to an increase in porosity and enhanced permeability. The dissolution phenomenon is particularly pronounced in acidic environments [12,13].

Some efforts have been made to meticulously elucidate the intricate features of black strata weathering through a comprehensive analysis encompassing chemical, mineralogical, and hydrological considerations [14,15]. Moreover, they succinctly expound upon the underlying mechanisms that give rise to disastrous consequences. The saturation index (SI) serves as a valuable metric for assessing the intricate interplay between water and rock, making it an indispensable tool for analyzing the dynamic nature of their interaction [16]. The utilization of the PHREEQC simulation software for chemical reactive transport processes during water–rock reactions has yielded promising results. The simulations have demonstrated the software's capability to effectively model and simulate the intricate geochemical reactions that occur under relatively simple chemical transport conditions. This success highlights the utility of PHREEQC as a powerful tool for studying and understanding complex water–rock interactions [17].

In this paper, the Xiaochatou landslide in the Wudongde reservoir area serves as a case study. Through comprehensive field geological surveys, the focus of the study lies in investigating the specific characteristics of water–rock interactions within the red-bed soft rock formation. Furthermore, an in-depth analysis of the instability mechanism of red-bed landslides under the influence of hydrochemical action is conducted. The research findings contribute significantly to the prevention and mitigation of geological disasters in the reservoir area. Consequently, they play a crucial role in ensuring the secure and

stable operation of the reservoir area, thereby safeguarding the surrounding environment and communities.

## 2. Site Description

The Xiaochatou landslide (Figure 1) is located on the right bank of the Jinsha River, approximately 2 km upstream of Malamo village, 98.3 km downstream of the Wudongde dam site, and 13.6 km upstream of Longchuan riverside town. The front edge of this landslide is close to the Jinsha River, at an elevation of around 1300 m. The gully has developed on both sides of the boundary. The maximum width of the front edge is about 570 m, which gradually decreases to 300 m, and its longitudinal length is approximately 700 m. The plane area is approximately 0.29 km$^2$. The thickness of the front edge and the trailing edge is 50–60 m, and the middle part is 110–120 m. Its volume is around $2400 \times 10^4$ m$^3$. The landform of this landslide area is gentle on the top and steep on the bottom, with an average gradient of 26° to 40°. A vertical ridge is about 30 m high at the trailing edge. The landslide body has a three-scale platform, with the 1080–1090 m elevation platform being approximately 30 m wide. The front edge is slightly upturned. Saggy terrain with banded distribution is located on the side of this landslide. The surface layer of this landslide is gravel soil containing gypsum debris; the middle layer is a thick block; and the bottom is mainly rock fragments. The underlying bed is sandstone, siltstone, and silty mudstone of the Cretaceous Matoushan Formation and Jiangdihe Formation. The Jiangdihe Formation contains gypsum lenses, and the rock layer tends to be northeast, with an inclination angle of 10° to 16°.

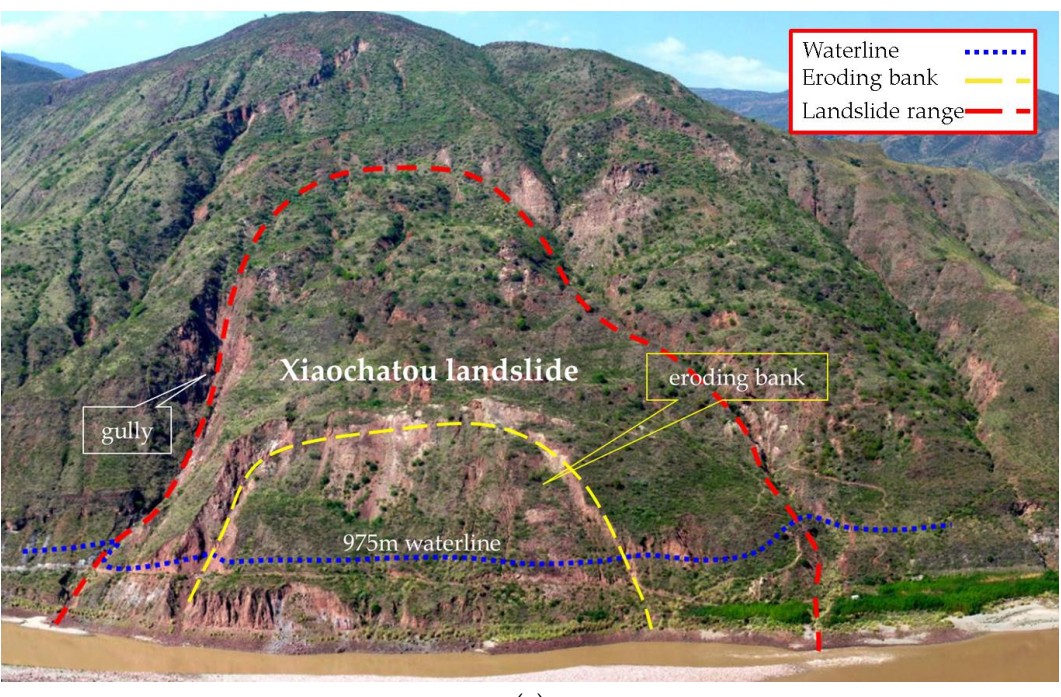

(**a**)

**Figure 1.** *Cont.*

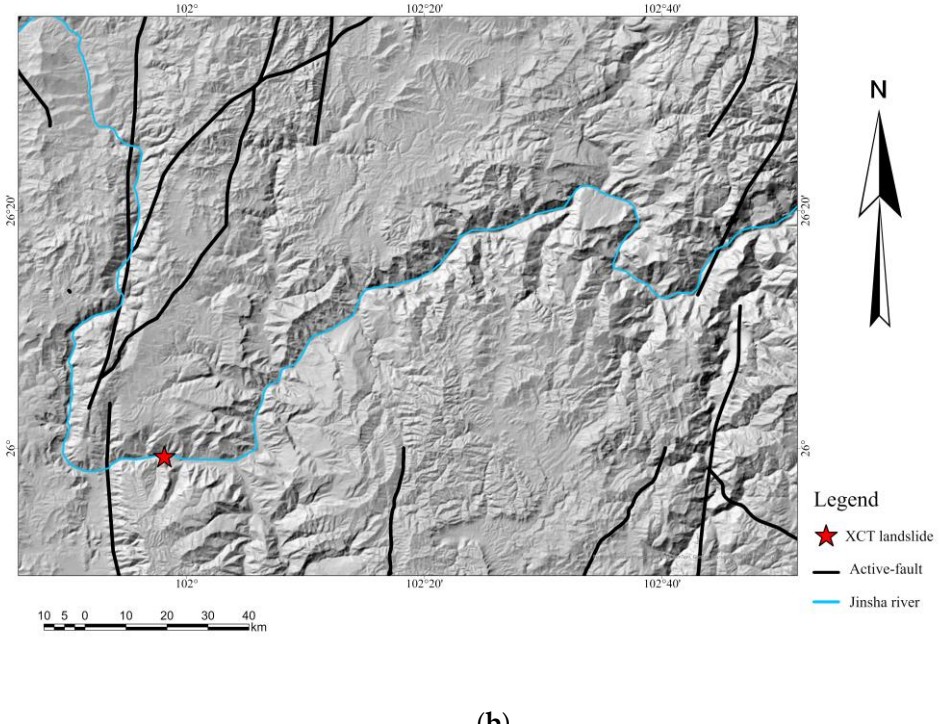

(**b**)

**Figure 1.** (**a**) The Xiaochatou landslide exhibits a distinctive landscape characterized by significant geological deformations and topographical alterations. (**b**) The study region, encompassing the Xiaochatou landslide, is located in a specific geographic location. It is typically described in terms of its geographical coordinates or by referencing nearby landmarks or boundaries. The location may be indicated using latitude and longitude coordinates or by specifying the region's relation to cities or geographical features.

## 3. Lab Analysis

The red-rock samples collected from the Xiaochatou landslide include conglomerate, sandstone, mudstone, calcareous cement, and soil samples from the bottom layer. The complete conglomerate rock and sandstone were prepared as cylindrical samples with a diameter of Φ50 mm and a height of 100 mm. Due to transportation constraints, the mudstone samples were small and were ready as cylindrical samples with a diameter of Φ25 mm and a height of 50 mm.

### 3.1. The Explanation of Samples and Key Lab Analysis

The photos of XCT-0 and XCT-0′ rock samples under the polarizing microscope are shown in Figure 2. The identification results of the rock samples are as follows:

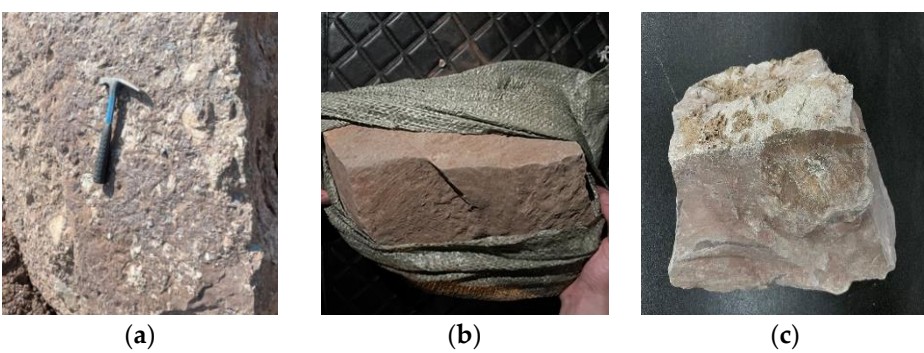

(**a**)            (**b**)            (**c**)

**Figure 2.** *Cont.*

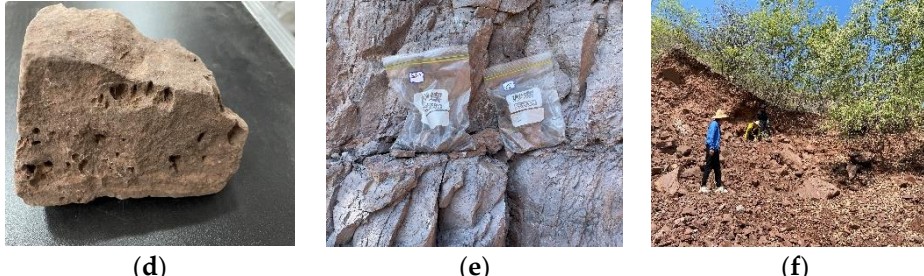

(**d**)                               (**e**)                               (**f**)

**Figure 2.** Landslide red-bed samples. (**a**) Conglomerate XCT-0, a calcareous sand conglomerate with gravels ranging from 2 mm to 50 mm, mostly round and sub-rounded shapes. Iron-rich sediment appears as reddish-brown dust mixed with rock debris. (**b**) Sandstone XCT-0′, primarily a fine-grained, calcareous sandstone. Debris is mostly angular to sub-angular, with particle sizes ranging between 0.05 and 0.2 mm, exhibiting poor roundness but good sorting. (**c**) Mudstone XCT-3, a fine-grained sedimentary rock composed of silt and clay-sized particles. (**d**) Dissolved sandstone XCT-2″, with some distinctive textures and structure, including enlarged pore spaces, dissolution vugs, and cavernous or pitted surfaces. (**e**) Calcium deposits XCT-2&2′. (**f**) Soil under pebbles XCT-3.

The gypsum in the red-bed rock mass of the reservoir area dissolves in groundwater and is collected in rock cracks and bedding by the water flow, as shown in Figure 3a. As the solution concentration increases, the gypsum precipitates in crystal form once it reaches dissolution equilibrium. The precipitated gypsum adheres to the rock mass surface or fills cracks between the original rock mass due to losing the gypsum mineral component in the original rock mass. This process destroys the structure of the red-bed rock mass, forming some dissolution holes, as shown in Figure 3b. This increases the porosity and enhances the permeability. The description of red-bed rock samples is shown in Table 1.

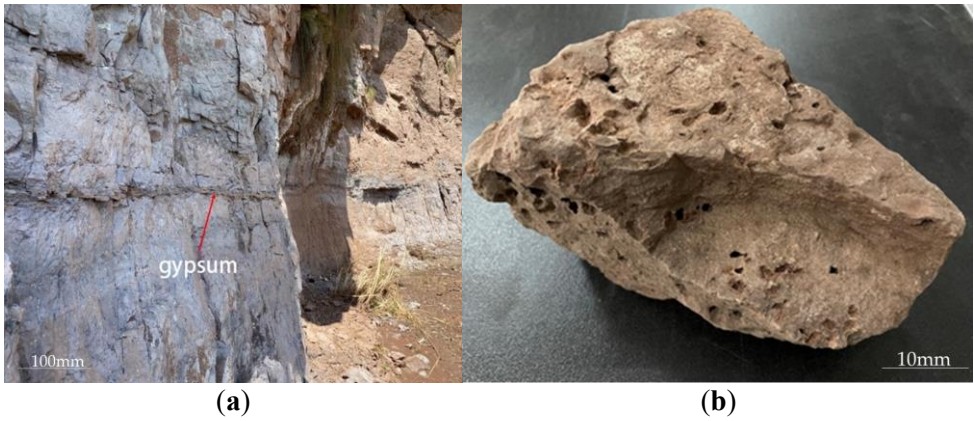

(**a**)                               (**b**)

**Figure 3.** Red-bed soft rock containing gypsum. (**a**) Gypsum layer filled in conglomerate cracks. (**b**) Dissolution cavities in conglomerate.

**Table 1.** Description of red-bed rock samples.

| Sites | Location (°) | Sample No. | Elevation | Description |
|---|---|---|---|---|
| | 25.989476, 101.969923 | XCT-0 | 967 | Conglomerate |
| | 25.989476, 101.969922 | XCT-0′ | 967 | Sandstone |
| Xiaochatou | 25.990128, 101.968432 | XCT-2 | 982 | Calcium deposits |
| landslide | 25.990128, 101.968433 | XCT-2′ | 971 | Calcium deposits |
| | 25.990128, 101968434 | XCT-2″ | 971 | Dissolved sandstone |
| | 25.990128, 101.968435 | XCT-3 | 971 | Mudstone |

### 3.2. Mineral Analysis and Identification

The mineral composition and proportion of the red-bed rock (soil) sample obtained by using X-ray diffractometer (XRD) are shown in Table 2.

**Table 2.** XRD mineral composition of red-bed rock.

| Sample [1] | Quartz | Calcite | Dolomite | Feldspar | Gypsum | Muscovite | Chlorite |
|---|---|---|---|---|---|---|---|
| XCT-0 | 37 | 12 | 34 | 8 | 8 | | 1 |
| XCT-0′ | 52 | 20 | | 13 | 10 | 4 | 1 |
| XCT-1 | 84 | 10 | | 6 | | | |
| XCT-2 | 7 | 93 | | | | | |
| XCT-2′ | 6 | 7 | | | 87 | | |
| XCT-2″ | 42 | 18 | | | | 40 | |
| XCT-3 | 30 | 18 | 36 | | | | 16 |

[1] The number in the table represents the percentage.

The rock sample XCT-0 is a calcareous sand conglomerate composed of several gravels and mixed interstitial materials. The gravels range from 2 mm to 50 mm and have mostly round and sub-rounded shapes. The composition of the conglomerate is diverse and includes four main types: (metamorphic) sandstone gravel, quartz schist gravel, dolomite gravel, and andesite basalt gravel. The interstitial materials are distributed among the gravel particles. The components include rock debris, quartz, feldspar, calcium, and iron-rich sediment. These include rock fragments, mostly between 0.15 and 2 mm in size, with a composition similar to gravel. The particle size of quartz and feldspar is between 0.05 and 1.00 mm. The quartz is primarily clear, while the feldspar is altered and cloudy. Calcite is present as granular particles with a particle size below 0.6 mm. These particles exhibit advanced white interference color and are stained red by the alizarin red reagent. The iron-rich sediment appears as reddish-brown dust mixed with rock debris between the sandy components.

Sample XCT-0′ is a fine-grained, calcareous sandstone composed of feldspar, quartz, and a significant amount of interstitial material (Figure 4). Debris in this sample is mostly angular to sub-angular. Particle sizes range primarily between 0.05 and 0.2 mm. The particles have poor roundness but good sorting. Most particles are in point-to-line contact. The debris consists primarily of quartz, feldspar, mica, and trace amounts of chlorite. The quartz in this sample is colorless with low protrusions and has a gray and gray-white interference color. The surface is unaltered and clean. Feldspar crystals exhibit dual characteristics, and both alkaline feldspar and plagioclase are visible. The surface is more turbid due to alteration. Mica, biotite, and muscovite are visible, mixed between felsic particles, and distribution is directional. Chlorite has a green-light-green interference color, often covered by its color or an abnormal interference color. Interstitial material is distributed between debris particles in a filling shape, and the components are primarily calcareous with trace amounts of siliceous and iron-muddy material. The calcium in this sample exhibits an advanced white interference color and foams when exposed to dilute hydrochloric acid, indicating the presence of calcite. The siliceous material is finely grained. The iron-muddy material is yellowish-brown to black-brown and reddish-brown, mainly in a granular form [18,19].

This section may be divided by subheadings. It should provide a concise and precise description of the experimental results, their interpretation, and the experimental conclusions that can be drawn.

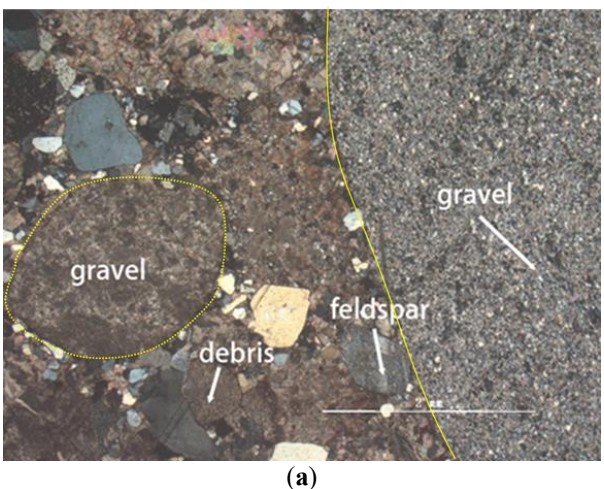

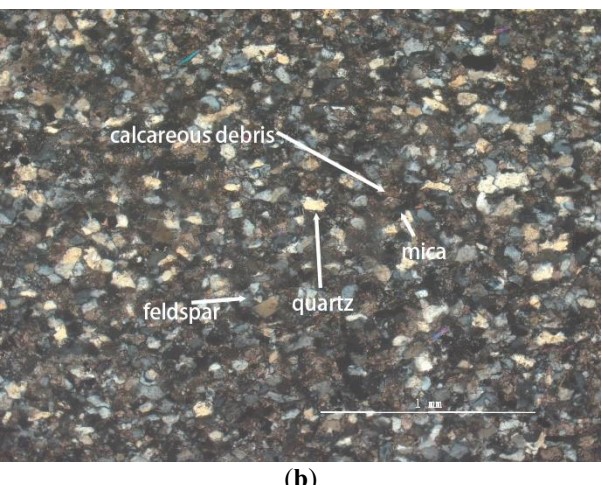

(**a**)                                                    (**b**)

**Figure 4.** Polarizing microscope photos of rock samples: (**a**) Polarizing microscope photos of XCT-0. (**b**) Polarizing microscope photos of XCT-0′.

*3.3. Water Immersion Dissolution Test*

3.3.1. The Variation Characteristics of Solution Ion Concentration during the Dissolution Process

In this dissolution study, five standard cylindrical samples of conglomerate and sandstone were weighed for their natural and oven-dried masses and placed in an empty container [2,3]. Subsequently, 1000 mL of deionized water was added to each sample. After 7, 14, 21, and 28 days, 10 mL of the soaking solution's supernatant was collected. We also observed the changes after immersion. The conglomerate sample showed relatively good durability during immersion, with minor surface cracks but overall structural stability. However, the mudstone sample showed significant alterations after immersion. Surface micro-cracks gradually formed, and thin layers detached, allowing easier water infiltration into the rock sample. This led to mineral hydration reactions, further compromising the internal structure of the mudstone. Therefore, compared to the conglomerate sample, the mudstone sample exhibited poorer durability in a moist environment, making it more susceptible to erosion and water damage. The concentrations of cations and anions in the dissolved solution were quantitatively analyzed using inductively coupled plasma emission spectrometry (ICP) and ion chromatography (IC). The result is shown in Figure 5.

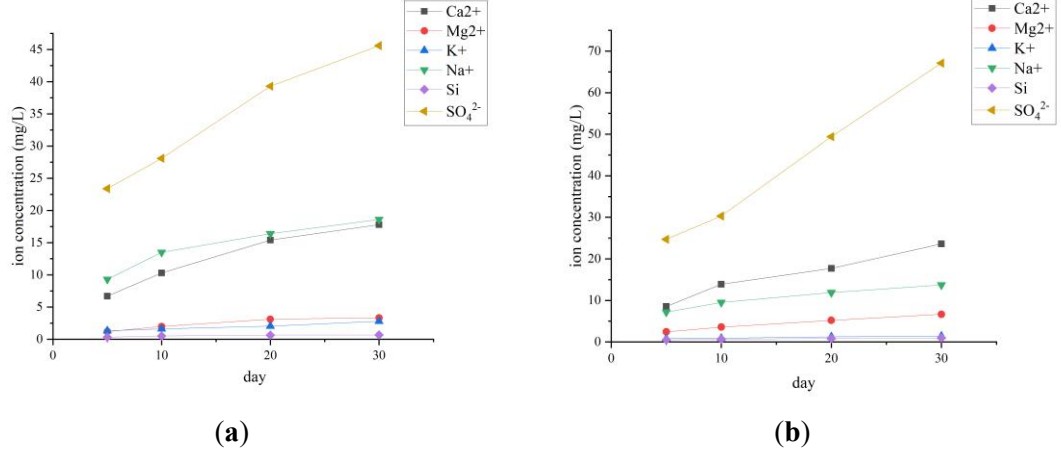

(**a**)                                                    (**b**)

**Figure 5.** Variation curve of ion concentration with time after immersion of rock samples. (**a**) Variation curve of ion concentration with time after immersion of conglomerate. (**b**) Variation curve of ion concentration with time after immersion of sandstone.

This indicates that the $SO_4^{2-}$ concentration is the highest in the soak solution of the conglomerate, and the highest cation concentrations are $Na^+$ followed by $Ca^{2+}$. The $SO_4^{2-}$ concentration has the highest rate of increase, followed by $Ca^{2+}$ and $Na^+$. Concentrations of other ions remained relatively stable. Similarly, the $SO_4^{2-}$ concentration is the highest in the immersion solution of sandstone, and the highest cation concentrations are $Ca^{2+}$, followed by $Na^+$ and $Mg^{2+}$. The $SO_4^{2-}$ concentration has the highest rate of increase, followed by $Ca^{2+}$ and $Na^+$. Concentrations of other ions remained relatively stable.

The mineral analysis revealed that the soluble minerals in the conglomerate and sandstone consist primarily of gypsum, calcareous cement, and chlorite. Due to the high concentrations of $Ca^{2+}$ and $SO_4^{2-}$ in the deionized water used in this analysis, gypsum is the primary mineral that dissolves. Gypsum is a soluble sedimentary rock commonly found in salt lake deposits. It has a chemical formula of $CaSO_4·2H_2O$ and belongs to the monoclinic crystal system [4]. Gypsum crystals are predominantly platy, compact, and fibrous and are more frequently found in red sandstone and shale. The following reaction can represent the hydrolysis of gypsum:

$$CaSO_4 \cdot 2H_2O \rightleftharpoons Ca^{2+} + SO_4^{2-} + H_2O \tag{1}$$

### 3.3.2. The Microstructure Characteristics before and after Immersion

In Figure 6a, a substantial quantity of fibrous and elongated gypsum cement is prominently dispersed across the surface of the rock mass specimen. Meanwhile, Figure 6c portrays the presence of chlorite crystals exhibiting a honeycomb-like configuration, reminiscent of delicate leaves adorned with slender hexagonal blades or plates. Prior to immersion, microscopic examination of these crystals revealed predominantly blocky and plate-like structures, characterized by smooth and level crystal planes, interconnected networks exhibiting close proximity, remarkable structural integrity, a limited presence of adhered clay minerals, and an absence of discernible signs of corrosion damage.

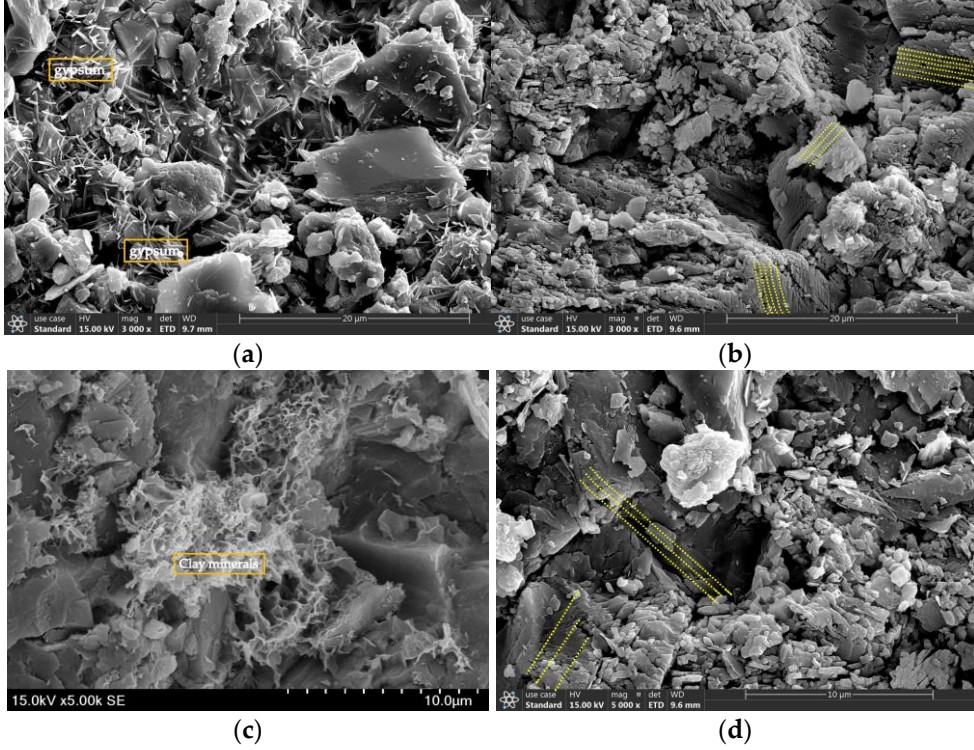

**Figure 6.** Conglomerate microstructure at different magnifications before and after immersion. (**a**) Microstructure before immersion (3000×). (**b**) Microstructure after immersion (3000×). (**c**) Microstructure before immersion (5000×). (**d**) Microstructure after immersion (5000×).

Figure 6b,d illustrate the alteration in the microscopic morphology of the rock sample subsequent to the soaking process, primarily attributed to water flow dissolution. The rock block undergoes fragmentation, resulting in the formation of smaller fragments, accompanied by distinct dissolution marks resembling knife-like incisions on the surface. Moreover, the overall integrity of the rock mass diminishes noticeably. Soluble minerals present in the rock, such as gypsum and chlorite, undergo dissolution and hydrolysis, giving rise to numerous dissolved pores. These dissolved pores significantly augment the contact area between the rock and water, facilitating further dissolution within the interior of the rock. Consequently, this dissolution process leads to the generation of fine particles as a product.

Figure 7a,c depict the initial state of the rock samples prior to immersion, showcasing stepped surfaces and sporadic distribution of gypsum. The crystals predominantly exhibit plate-like and elongated strip structures, with a few discernible small cracks. In contrast, Figure 7b,d provide a look into the altered microscopic morphology of the rock sample subsequent to the dissolution of gypsum minerals in deionized water, resulting in the formation of dissolution holes. The fine particles present on the surface undergo either exfoliation or dissolution, leading to a rough and loosely structured arrangement of the crystal planes. Following the dissolution process, the structure assumes a flaky configuration, while the micro-cracks gradually extend and widen, further compromising the integrity of the rock sample.

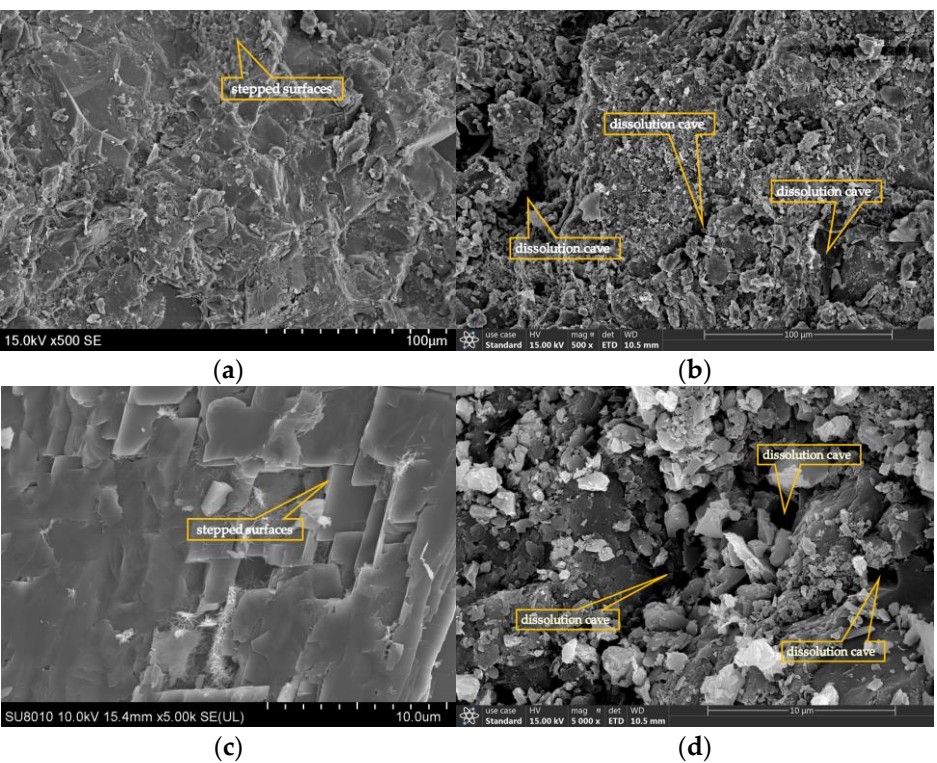

**Figure 7.** Sandstone microstructure at different magnifications before and after immersion. (**a**) Microstructure before immersion (3000×). (**b**) Microstructure after immersion (3000×). (**c**) Microstructure before immersion (5000×). (**d**) Microstructure after immersion (5000×).

## 4. Numerical Analysis of Dissolution Effect for Red Beds

To analyze the process of dissolution, we conducted an equilibrium process simulation using the hydrogeochemical software PHREEQC, focusing on conglomerate and sandstone samples from the Xiaochatou landslide [17,20–23]. We simulated the dissolution and precipitation of red-bed minerals using the saturation index method [24–27].

*4.1. Saturation Index*

The saturation index is a vital mineral parameter based on the principle of thermodynamics. According to the SI, we can determine the reaction state between water and minerals [28].

For the reaction:

$$aA + bB = cC + dD \tag{2}$$

The AP (activity product) is expressed as:

$$AP = [C]^c[D]^d / [A]^a[B]^b \tag{3}$$

If all components are ions, they can be represented by IAP (ion activity product). When the reaction reaches equilibrium, we assumed there are a′ mol A, b′ mol B, c′ mol C, and d′ mol D. According to the law of mass action, the activity product K is as follows:

$$K = [C]^{c'}[D]^{d'} / [A]^{a'}[B]^{b'} \tag{4}$$

The expression of the saturation index SI is:

$$SI = \lg \frac{IAP}{K} \tag{5}$$

The mineral is saturated or supersaturated when the SI is 0 or greater than 0. This mineral will tend to stop dissolving or even precipitate. When the SI is less than 0, the mineral is unsaturated in groundwater. When it still exists in the water environment, it will tend to dissolve and gradually become saturated.

When the saturation index (SI) of a mineral is 0 or greater than 0, it indicates that the mineral is either saturated or supersaturated in the given environment. In such a state, the mineral is prone to cease dissolving or even precipitate out of the solution. The presence of excess minerals beyond saturation levels promotes the formation of solid precipitates.

Conversely, when the SI is less than 0, it signifies that the mineral is unsaturated in the groundwater. In this scenario, if the mineral persists in the water environment, it will tend to dissolve gradually. Over time, as dissolution continues, the mineral concentration in the water may increase, eventually reaching a point of saturation.

*4.2. The Simulation of the Process of Dissolution*

The saturation index method calculates the dissolution of red-bed minerals, and the dissolution and precipitation states of red-bed conglomerate and sandstone mineral components in such an environmental aqueous solution can be determined. This way, we can predict the chemical weathering trend of the red bed. This paper mainly analyzes the variation in the SI of these rock-forming minerals, which greatly influence rock mass structure, such as albite, gypsum, K-mica, illite, and kaolinite. Table 3 shows the ion concentration analysis results of samples in the layer area.

**Table 3.** Water quality analysis unit: mg/L.

| Rock Sample | Time (Day) | Al | $Ca^{2+}$ | $Mg^{2+}$ | $K^+$ | $Na^+$ | Si | $SO_4^2$ |
|---|---|---|---|---|---|---|---|---|
| conglomerate | 5 | 0.04 | 6.70 | 1.19 | 1.32 | 9.30 | 0.34 | 23.4 |
| | 10 | 0.03 | 10.30 | 1.99 | 1.63 | 13.50 | 0.52 | 28.1 |
| | 20 | 0.04 | 15.4 | 3.12 | 2.06 | 16.4 | 0.650 | 39.3 |
| | 30 | 0.03 | 17.8 | 3.33 | 2.81 | 18.6 | 0.688 | 45.6 |
| sandstone | 5 | 0.07 | 8.54 | 2.43 | 0.83 | 7.19 | 0.51 | 24.7 |
| | 10 | 0.05 | 13.90 | 3.58 | 0.92 | 9.52 | 0.69 | 30.3 |
| | 20 | 0.07 | 17.7 | 5.21 | 1.24 | 11.9 | 0.859 | 49.4 |
| | 30 | 0.06 | 23.6 | 6.67 | 1.40 | 13.7 | 0.978 | 67.1 |

This paper used the PHREEQC compiled by the United States Geological Survey (USGS) to analyze the above water samples. To obtain the SI value that can better reflect the actual water–rock interaction state, the calculation method of this software is as follows: First of all, the various forms of dissolution groups in the water–rock system were solved. Then, the concentration of free ions was obtained by calculating various complexes. This step could also calibrate the ionic strength and determine ion activity more accurately. Finally, the SI value was obtained [16,29]. The conditions of water samples (temperature, pH, ion concentration, etc.) were input into the PHREEQC. The thermodynamic database automatically installed with PHREEQC can establish the corresponding equations. The equations were solved by the Newton–Raphson iteration method. Considering the influence of the complex, the SI value of each mineral in the water sample was finally calculated.

Based on the observations presented in Figure 8, it is evident that the saturation index (SI) values for kaolinite and K-mica consistently exceed 0 in both the conglomerate and sandstone samples. On the other hand, the SI values for the other minerals in the conglomerate sample remain below 0. Similarly, in the sandstone sample, only kaolinite demonstrates an SI value above 0.

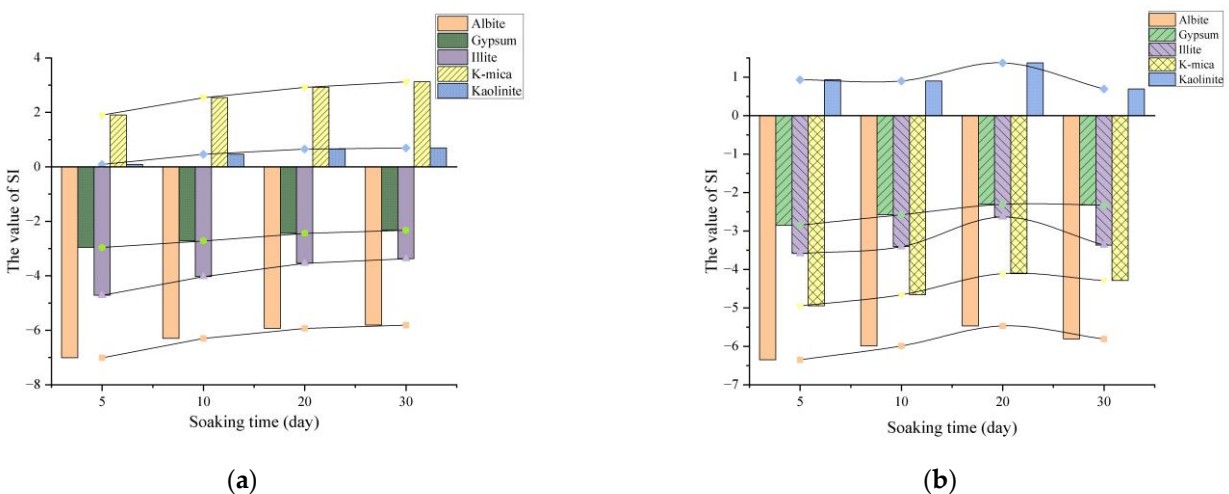

(**a**)                                                                 (**b**)

**Figure 8.** (**a**) Trend chart of SI of the conglomerate sample with soaking time. (**b**) Trend chart of SI of sandstone sample with soaking time.

These findings suggest that illite, gypsum, and albite do not separate in or precipitate out of the conglomerate samples. Instead, minerals such as kaolinite and K-mica gradually undergo precipitation over time. In contrast, the sandstone sample exhibits a different pattern. Throughout the entire process, only kaolinite is observed to precipitate, while the other minerals do not exhibit significant changes in their SI values.

These observations provide valuable insights into the mineralogical behavior within the conglomerate and sandstone samples, highlighting the selective precipitation of specific minerals over others under the given conditions.

### 4.3. The Sensitivity Analysis of Mineral Dissolution and Precipitation

Interestingly, certain environmental conditions play a significant role in the process of dissolution and precipitation. Therefore, it was imperative to conduct a sensitivity analysis on various parameters during the numerical analysis [30–32].

A sensitivity analysis serves several purposes in research. Firstly, it helps identify the most critical parameters that significantly affect the model's output. This understanding is crucial for focusing resources and efforts on the most influential factors. Secondly, a sensitivity analysis allows us to assess how different combinations of parameters impact the model's output process, providing insights into complex interactions and potential synergies or trade-offs. Lastly, a sensitivity analysis helps identify parameters that have a

minimal impact on the model's output, allowing for a reduction in computational burden and uncertainty associated with those parameters.

In the referenced paper, the dissolution and precipitation of gypsum are recognized as playing a prominent role in the Xiaochatou landslide [33–36]. Consequently, the paper conducted a sensitivity analysis specifically focused on the dissolution of gypsum in water samples obtained from the Jinsha River. The results of the water quality analysis, including the relevant data, are presented in Table 4. These findings are crucial for understanding the behavior of gypsum in the context of the landslide and contribute to a comprehensive understanding of the factors influencing the landslide dynamics.

**Table 4.** Water quality analysis of JSJ-1 unit: mg/L.

| Sample | $Ca^2$ | $Mg^{2+}$ | $K^+$ | $Na^+$ | $PO_4^2$ | $HCO_3$ | $SO_4^2$ | pH |
|---|---|---|---|---|---|---|---|---|
| JSJ-1 | 53.5 | 17.7 | 2.71 | 48.2 | <3 | 184 | 62.9 | 7.65 |

Using the EQUILIBRIUM_PHASES in PHREEQC, some parameters were set as the environmental variables, and others remained unchanged [37]. We considered six parameters in this paper: the ion concentrations of $Ca^{2+}$, $Mg^{2+}$, $Na^+$, $HCO_3^-$, $SO_4^{2-}$, and the value of pH. We assumed this dissolution process happened in 1 kg of a JSJ-1 water sample. The dissolution and precipitation capacity of minerals was characterized by the amount of minerals dissolved and liberated and the value of SI during the dissolution equilibrium. Figure 9 plots the results on the graph.

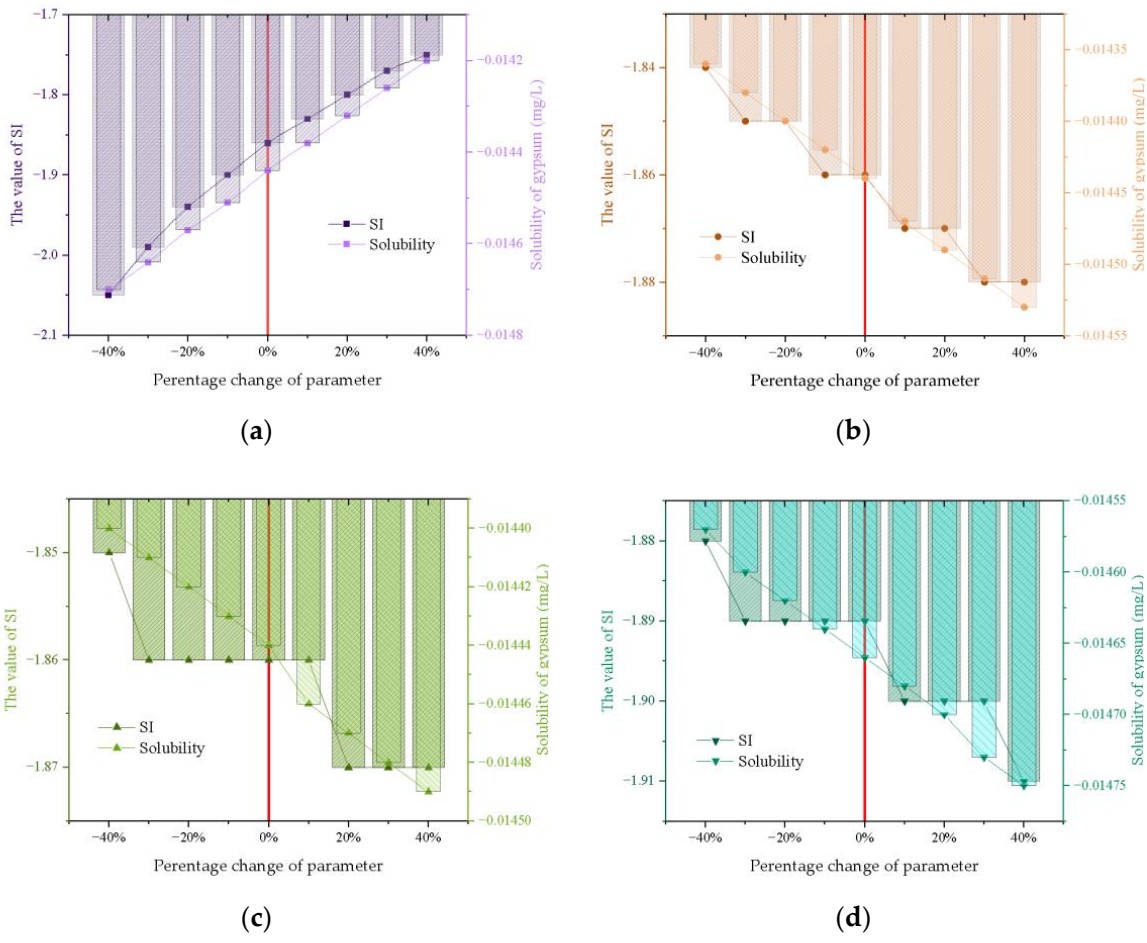

**Figure 9.** *Cont.*

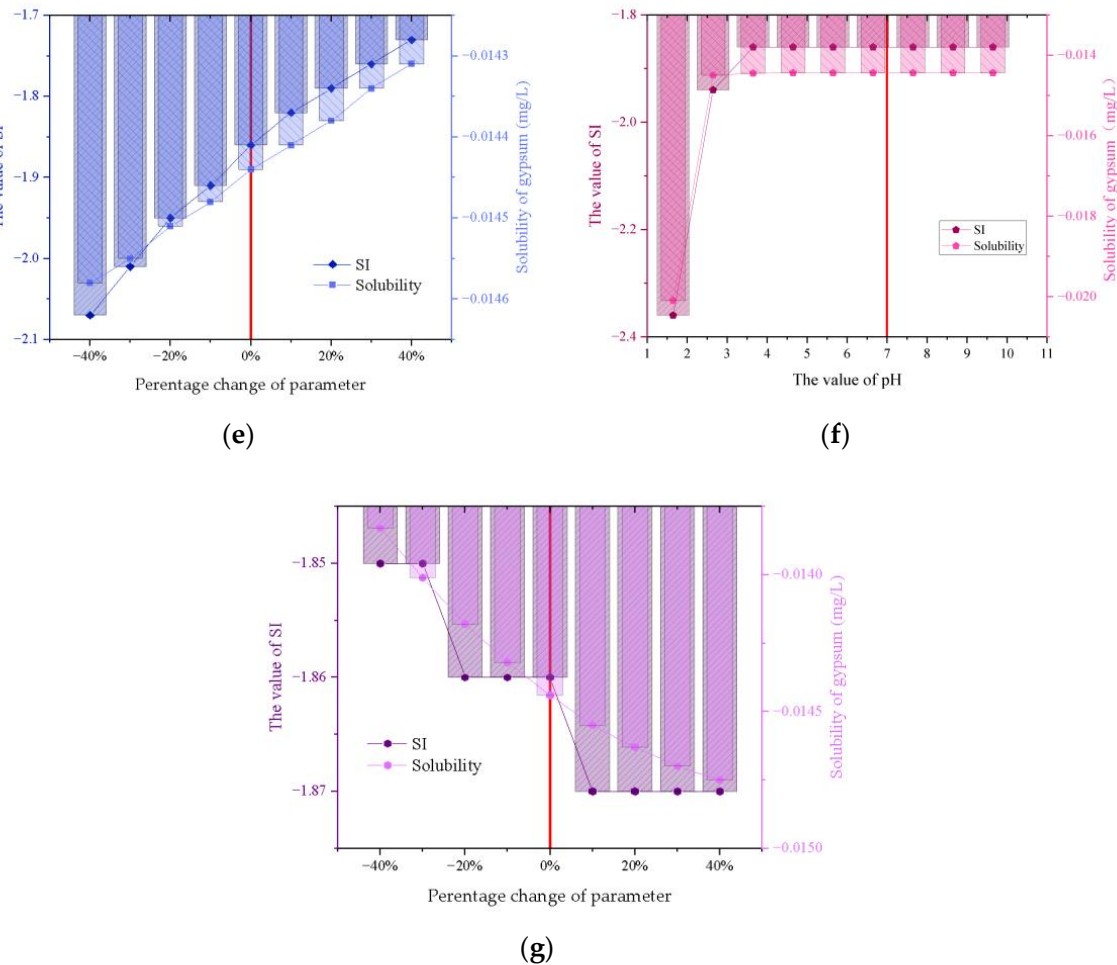

**Figure 9.** The curve of dissolution and precipitation ability of gypsum with different parameters change. Figure 9g plot shows the superposition process of the above parameters. (**a**) $Ca^{2+}$; (**b**) $Mg^{2+}$; (**c**) $Na^+$; (**d**) $HCO_3$; (**e**) $SO_4^{2-}$; (**f**) pH; (**g**) SUM.

Then, a sensitivity analysis was performed [38–41]. In this paper, the sensitivity function was used to evaluate gypsum dissolution. The sensitivity function can be available in closed form or given only as a numerical procedure to compute y given by x. Typically, the output y is a scalar variable that "summarizes" the wide range of variables (often time series, possibly spatially distributed) provided by the simulation. The sensitivity function of each parameter can be obtained from Figure 9 by the least square method. To rank the influence of each parameter, the sensitivity indices that the sensitivity function can define are often employed in a sensitivity analysis and important measures. These synthetic indices quantify the relative contribution to output uncertainty from each input. A sensitivity index of zero means that the associated information is non-influential, while the higher the index, the more influential the intake. In this case, the slope of the linear fitting of each parameter and the range of change were considered for such a sensitivity index. The sensitivity ranking of each parameter is given in Table 5. The sensitivity ranking from large to minor is $SO_4^{2-}$, pH, $Ca^{2+}$, $Mg^{2+}$, $HCO^{3-}$, and $Na^+$.

**Table 5.** The sensitivity function and sensitivity index of each parameter.

| Parameter | Function | Index |
|:---:|:---:|:---:|
| $Ca^{2+}$ | y = 0.36x − 1.87 | 0.36 |
| $Mg^{2+}$ | y = −0.05x − 1.86 | 0.05 |
| $Na^+$ | y = −0.02x − 1.86 | 0.02 |

**Table 5.** *Cont.*

| Parameter | Function | Index |
|:---:|:---:|:---:|
| $HCO_3^-$ | y = −0.03x − 1.89 | 0.03 |
| $SO_4^{2-}$ | y = 0.42x − 1.87 | 0.42 |
| pH | y = 0.4x − 2.13 | 0.40 |
| SUM | y = −0.03x − 1.86 | 0.03 |

## 5. Discussion

The water immersion test results indicate that sandstone is more susceptible to dissolution by water [19]. Additionally, the experiment on the wave velocity of mudstone during the dry–wet process reveals that the strength attenuation of mudstone is significantly reduced [42,43]. Furthermore, we examined the dissolution saturation of gypsum in the water of the Jinsha River. Based on the results above, we can have the following discussions:

### 5.1. Discussion of the Lab Analysis

The composition of both XCT-0 and XCT-0′ revealed by XRD is further elucidated when combined with the SEM images of these two samples. The result of the test shows a presence of fibrous and elongated gypsum cement, which is prominently dispersed across the rock mass surface. Additionally, it is observed that certain clay minerals, such as gypsum, dissolve under water.

The soluble minerals within the conglomerate and sandstone are predominantly composed of gypsum, calcareous cement, and chlorite. According to the high concentrations of $Ca^{2+}$ and $SO_4^{2-}$ ions in the deionized water utilized in the lab analysis, gypsum is identified as the primary mineral that undergoes dissolution.

### 5.2. Discussion of the Numerical Analysis

The simulation of sandstone and conglomerate dissolution indicates that chemical weathering is a significant factor in forming cracks on the rock surface during dissolution [25]. Furthermore, by considering the changes in the $Ca^{2+}$ and $SO_4^{2-}$ ion concentrations and SI values, we can conclude that gypsum is the primary dissolved substance during the dissolution process.

A qualitative analysis of gypsum dissolution is necessary. Nevertheless, the mechanism of how dissolution occurs under various environmental conditions is too complex to explain clearly. Therefore, we selected some typical parameters for a sensitivity analysis. While the equation for gypsum is suitable for interpreting the high sensitivity of $SO_4^{2-}$ and $Ca^{2+}$, it cannot provide a reasonable explanation for the variation phenomenon when the concentrations of $Mg^{2+}$, $HCO_3^-$, and $Na^+$ change. The slight effect of changes in the ion concentrations on the SI value can be explained by the pH value, which shifts the reaction equilibrium [24,29]. The pH and SI curves demonstrate that the SI value increases in an acidic environment but approaches 0 in a neutral environment. This explains the variation phenomenon when the concentrations of $Mg^{2+}$, $HCO_3^-$, and $Na^+$ change. The SUM plot illustrates the result when the above parameters change together. This figure demonstrates that the influence of each parameter on the dissolution effect is not simply additive, explaining that this reaction requires an understanding of the movement of reaction equilibrium. The dissolution of gypsum in rock samples predominantly contributes to the overall mineral depletion. It has varying sensitivity to changes in environmental parameters, resulting in decreased mechanical properties of the rock samples.

### 5.3. Discussion of the Mechanism of Xiaochatou Landslide

The red-layer landslide in the reservoir area consists of thick sandstone interbedded with thin mudstone. The sandstone, characterized as hard rock, contains gravel at its base, while the mudstone is classified as soft rock [10,29,44,45]. The sandstone, which contains soluble minerals such as gypsum, is highly permeable and acts as an aquifer, while the

mudstone is impermeable. Water infiltrating from the ancient landslide-dammed lake dissolves gypsum, clay minerals, and calcium cementation in the sandstone, which are carried away by the water flow, forming numerous dissolution pores that further enhance permeability. Tensile cracks perpendicular to the sliding surface form inside the sandstone near the locking or misalignment points of the sliding surface due to the concentration of tensile stress. Cracks facilitate surface water infiltration, which flows along the cracks and converges at the interface between sandstone and mudstone. The water further infiltrates the slope, softening the mudstone and reducing its strength. The upper sandstone fracture surface reforms and continuously expands upward.

## 6. Conclusions

(1) The sandstone conglomerates in the red beds of the Wudongde reservoir area contain soluble minerals such as gypsum and chlorite. When the rock samples are immersed in water, the dissolved ions are mainly $SO_4^{2-}$, $Ca^{2+}$, and $Na^+$, and the primary mineral dissolved is gypsum. The microstructure of the rock changes after immersion, with easily dissolved pores being generated, which increases the permeability of the rock mass.

(2) The dissolution of sandstone and conglomerate is affected by chemical weathering, which forms cracks on the rock surface. Gypsum is the primary substance dissolved during the process. The dissolution mechanism is intricate under diverse environmental conditions, and the effect of changes in ion concentration on the SI value is explained by the pH value that shifts the reaction equilibrium. The SUM plot shows that the impact of each parameter on the dissolution is not simply additive. Gypsum dissolution accounts for most of the total dissolution and decreases the mechanical properties of rock samples.

(3) The deformation pattern manifested by the landslide encompasses the emergence of slip-compression-induced tensile fractures within the strata characterized by a shallow dip angle. Within this framework, the presence of water from the Jinsha River, permeating the mudstone and facilitating gypsum dissolution, assumes a pivotal role as the regulating layer for slip control. Consequently, the sandstone undergoes internal structural changes, characterized by the formation of vertically elongated fissures that exhibit a narrow apex and a broader base. Throughout the progressive deformation, these fissures persistently propagate and extend vertically until the stability threshold of the slope rock mass is surpassed, ultimately leading to its catastrophic collapse.

**Author Contributions:** Conceptualization, S.Z.; Methodology, C.Y.; Software, S.Z.; Investigation, C.Y.; Resources, J.L. and H.L.; Writing—original draft, C.Y.; Writing—review and editing, C.Y.; Supervision, Y.S. and J.D.; Project administration, J.D.; Funding acquisition, H.L. All authors have read and agreed to the published version of the manuscript.

**Funding:** This study was financially supported by the Science and Technology Major Project of Tibetan Autonomous Region of China (XZ202201ZD0003G) and the National Natural Science Foundation of China (U22A20601).

**Data Availability Statement:** The original contributions presented in the study are included in the article.

**Conflicts of Interest:** The authors declare no conflicts of interest. Author Yan Shi was employed by Three Gorges Geotechnical Engineering Co., Ltd. Author Jingmin Liu was employed by Power China Huadong Engineering Corporation Limited. The remaining authors declare that the research was conducted in the absence of any commercial or financial relationships that could be construed as a potential conflict of interest.

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
