# Peer review of "The Mechanism of Mineral Dissolution on the Development of Red-Bed Landslides in the Wudongde Reservoir Region"

_minerals, doi:10.3390/min14010115_

Round 1

Reviewer 1 Report

Comments and Suggestions for Authors

This manuscript investigated the mechanism of red bed landslide via hydrochemistry simulation. A typical landslide was selected to analyze the relationship between mineral dissolution and landslide development. Generally, this is an interesting article providing a case study for red bed landslides and soaking process of gypsum in the red layer. I suggest a moderate revision before its publication. Some modification should be made on this article.

1.      The abstract needs some more quantitative description, in particular, by addressing the “development” in the title.

2.      Line 91-93. The sentence contained too many commas. Please modify it.

3.      Section 2. Please make some explanation that how typical this landslide is and why it was chosen to be illustrated here.

4.      Line 246. Please state why it is difficult and challenging. This is important to lead to your motivation of this study. Please make a proper expression on the key necessity of your job.

5.      Figure 7 and 8. Texts in the legend are tiny.

6.      The sections of Introduction or Study area needs to be improved about the description of the regional geological background. This is important to leading to your own task of this study.

7.      Line 183. Please provide some necessary reference to support your description.

8.      Line 251. Please provide some necessary reference to the saturation index.

9.      Line 372. As presented by the authors, the gypsum dissolution plays an important part in the erosion process of sandstone and mudstone layer. Please add some explanations about what controls these patterns, and some discussions on what is the difference in landslide development between the study area. For now, few discussions are presented in the manuscript.

10.   Figure 1a. The word appears visually distorted and presents challenges in its identification.

11.   Figure 1b. It is hard to recognize the yellow and orange polygons of DEM in the figure. Please make some adjustments.

12.   Line 417. The sentence is too long with many semicolons and commas. Please rewrite them.

13.   Figure 9c. The Texts inserted in the figure are difficult to identify.

14.   The sentences and grammars of the thorough manuscript could be further polished and improved for the readability.

Comments on the Quality of English Language

Minor editing of English language is required.

Author Response

Thank you for pointing this out.  The respond letter is as follows:

Reviewer 2 Report

Comments and Suggestions for Authors

kindly see the attached annotated file for detailed comments. Some of the key suggestions are as follows:

Abstract section

1. clarify what is meant by the red bed...if international readers are going through your work, they are not known by "Red Bed". first, in the title it is termed as Red Bed landslides and in abstract section, it is assoicated with reservoir. vague statement.

2. what is meant by reservoir area and it is termed as a reservoir with respect to water resources or petroleum resources? secondly, if it is associated with petroleum, specify that it is conventional or unconventional.

3. the application and significance of the study are not well-defined. revise the statement accordingly.

Keywords:

4. add the keywords: Scanning electron microscopy (SEM); siliciclastic rocks and evaporites

Introduction:

5. describe the significance of Red Beds. and their regional or global presence.

6. what are the key engineering characteristics of red beds that result in natural disasters and landslide movement?

site description section:

7. The site description is written without referring to the literature. include the key references regarding the landslide location and description.

8. fig 1 should be labeled with font color with contrast for better visibility. increase the resolution of the figure and redraw the key features of the figure to explain the components of the landslide.

section 3. Characteristics............red beds

9. a section related to the explanation of samples/dataset and key lab analysis should be added to the manuscript.

10. Revise the section "3. The characteristics of the reaction of water-rock interaction in red bed rock" as "Lab analysis" 

11.  Fig.2a: The images should not contain the shadow of the person or any object. replace the figure with the better one

12. Fig. 2 a-e: the scale of the rock sample could be added for better presentation.

13. Figure 3a: what is meant by debris? it should be properly explained and the gravel could not be smaller than feldspar grain. check the labeling of the photomicroscopy and label them correctly.

14. Figure 5: include the scale of the images and label them with all the features.

 15. Figure 7: label the SEM images with details and use a font color that is visible in the figure.

section 4: numerical analysis.......red beds

16. Fig. 9: The axis label is not clear in the graphs. redraw the graphs or increase the resolution.

discussion section

17. include a subsection in the discussion paper to discuss the results of mineral analysis and lab analysis.

general comment: reduce the similarity index. and don't use AI writing tools as it is 15% in your manuscript.

Author Response

Thank you for pointing this out. 

Reviewer 3 Report

Comments and Suggestions for Authors

Paper well written, must it will be .

It is suggested as the porosity and permeability are major findings of the work and are controls on the landslide stability that they should be determined by direct tests before and after the water immersion tests in order to support the conclusions.

Durability of the material should be described using the classifications given in the literature. It is not describe how much is the disintegration (slaking) of the material.

Improve the quality of figure 1b.

Comments on the Quality of English Language

Well written a final revision is need.

Author Response

Thank you for pointing this out. 

Round 2

Reviewer 2 Report

Comments and Suggestions for Authors

Fig. 2: Figure 2a is vertically exaggerated and the aspect ratio is not maintained. also label the figure and explain. 

Figure 3a must contain the scale of the image. include the scale of the field image.

Figure 6 and figure 7 with white text labeling. change the text color for better visibility.

include a table to show the number of rock samples included in this study

Author Response

Thank you for pointing this out.

Reviewer 3 Report

Comments and Suggestions for Authors

The suggestions recommended before were not done. The subjects concerning porosity, permeability and slaking of the material were not improved.

Author Response

Thank you for pointing this out.
